# Investigation on the Impact of Trailing Edge Stacking Styles on Hydraulic Performance in the Multistage Submersible Pump Space Diffuser

Hui Zhang [1,2], Puyu Cao [1,*], Dan Ni [3], Xuran Gong [1], Bo He [4] and Rui Zhu [5]

1   National Research Center of Pumps, Jiangsu University, Zhenjiang 212013, China;
    2112011002@stmail.ujs.edu.cn (H.Z.); 2221811015@stmail.ujs.edu.cn (X.G.)
2   Department of Automotive Engineering, Zhenjiang Technician College Jiangsu Province,
    Zhenjiang 212013, China
3   School of Energy and Power Engineering, Jiangsu University, Zhenjiang 212013, China; nidan@ujs.edu.cn
4   Shanxi Wofeng Fluid Technology Co., Ltd., Xi'an 710100, China; 7037.hebo@163.com
5   School of Energy and Power Engineering, Xi'an Jiaotong University, Xi'an 710049, China;
    ruizhu@stu.xjtu.edu.cn
*   Correspondence: mafatu1988@ujs.edu.cn; Tel.: +86-199-0929-2863

**Abstract:** To investigate the effect of the different wrap angles from the hub to the shroud surface in the space diffuser (i.e., the trailing edge stacking style) on the principle of corner separation vortex flow, a numerical simulation method has been conducted in a multistage submersible pump. Building a linear equation on the profile line of the diffuser trailing edge to optimize the wrap angle on every spanwise from the hub to the shroud, and the mapping response relationship between the wrap angle difference and the hydraulic performance in the space diffuser has been analyzed. Under the variable wrap angle difference ($\Delta\phi = \phi_{hub} - \phi_{shroud}$), the secondary flows in different directions, non-uniformity, diffuser efficiency, and pressure recovery are compared. The positive wrap angle difference (i.e., the shroud wrap angle is smaller than the hub one) improves the strength of the secondary flow and partly corner separation vortex in the diffuser, so the hydraulic characteristic of positive cases is better than the negative wrap angle difference. Moreover, in scheme A (in which the hub wrap angle is constant and the shroud wrap angle decreasing), the transversal secondary flow has been weakened, the low-energy fluid located in the corner has been suppressed, the extensional secondary flow has been increased, the diffuser hydraulic performance has been improved, and unidirectionally increases with the wrap angle difference increasing. When the shroud wrap angle is constant, the extensional secondary flow has been enhanced by the increasing hub wrap angle. Meanwhile, the increasing extensional secondary flow has been countered by the deteriorating extensional flow at the diffuser inlet and transversal secondary flow, the diffuser hydraulic performance increases and then decreases as the wrap angle difference increases, with an optimal wrap angle difference is about $20°$.

**Keywords:** space diffuser; trailing edge; stacking styles; wrap angle; optimization method

## 1. Introduction

Multistage submersible pumps are widely used in agricultural irrigation, water supply, petroleum transport, and drainage in isolated mountain regions [1–3], etc. A variable working environment requires higher stability of pump performance. Increasing the single-stage pump head can improve the reliability of the pump operation and reduce the manufacturing cost. The multistage submersible pump space diffuser transforms the higher velocity kinetic energy from the impeller outlet into hydrostatic energy and conveys it to the next stage impeller inlet or pump outlet pipe. Relevant studies have shown that the space diffuser hydraulic loss accounts for about 40~50% of the total hydraulic loss of

the pump [4], which has a great influence on the pump's performance [5]. Therefore, the optimization of the space diffuser is necessary to improve the single-stage head and the overall performance of the multistage submersible pump.

Several researchers have studied the mechanism of the hydraulic loss of the space diffuser in recent years. Xu et al. [6] applied the Liutex vortex identification method to analyze the blade passage vortex influence on hydraulic loss. They investigated that the amount of blade passage vortices dramatically increases the hydraulic loss of the draft tube. The hydraulic loss in the blade passage accounts for 32.6% of the total channel hydraulic loss at the designed flow rate. Qin et al. [7] regarded hydraulic loss as the interaction of the dissipation effect with the transportation effect. Goto et al. [8] discovered a large-scale separation vortex in the corner of the "hub-suction surface" in the space diffuser of a pump, as shown in Figure 1, and illustrated that the separation vortex is the source of the hydraulic loss of the diffuser. Scillitoe et al. [9] applied the large eddy simulation method to confirm that the loss of the compressor is dominated by the corner separation vortex and the diffuser outflow wake. The research by Gbadebo et al. [10] proposed that the compressor optimal method should focus on limiting the three-dimensional flow separation in the corner region.

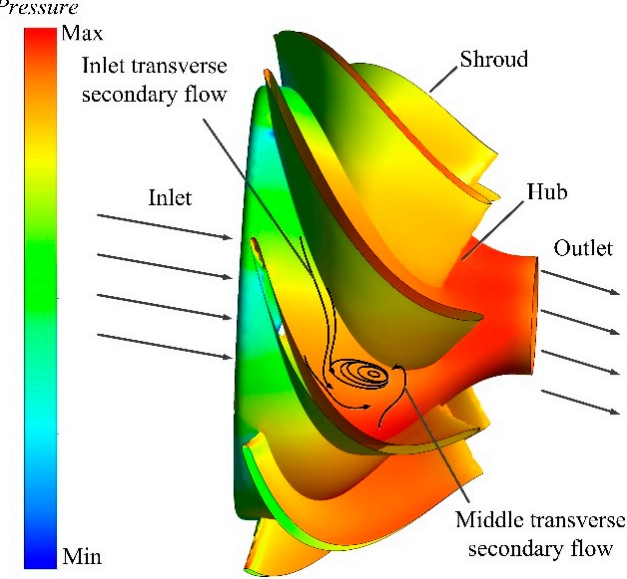

**Figure 1.** Corner separation vortex of the space diffuser [8].

In terms of the methods of suppressing the space diffuser corner separation vortex, Goto et al. [11,12] mainly adopted the inverse design method to design two types of blades, i.e., shroud back-loaded and hub forward-loaded blades, for suppressing the corner separation vortexes, the flow was assumed to be irrotational and away from the actual flow conditions. Zhao et al. [13] and Zhang et al. [14] mainly adopted the method of combining simulation calculation and experiments to analyze the influence of different wrap angles and the number of blades on pump performances when the wrap angle difference is zero, but the inner flow characteristics in the space diffuser were not clear.

In the field of aerodynamic and turbine machinery design, cascade stacking design in 3D [15–17] was considered to improve performance. Rosic et al. [18] compared three types of static blades with low aspect ratio turbines and found that the positive wrap angle difference affects the load distribution of blades which could suppress the leakage flow on the hub surface. Ma et al. and Liu et al. [19,20] showed that large inlet blade lean can inhibit separation flow and improve turbine efficiency using a multi-objective optimization design method. The research by Razavi et al. [21] and He et al. [22] reported that the optimal stacking angle of the transonic rotor could improve efficiency and stability. Jang et al. [23,24] showed that blade lean design can improve the adiabatic efficiency of transonic

axial flow compressors. At present, the cascade three-dimensional stacking design method is mostly applied to compressors and turbines [25,26] but is rarely applied to the pump, especially the space diffuser.

Based on the method of the aerodynamic machinery, the objective is to improve the hydraulic performance of the space diffuser by the stacking design method in this paper. A linear function has been established to control the space diffuser trailing edge structure under the condition of the inlet and outlet blade angle of the space diffuser unchanged. The relationship between the linear function and the wrap angle of the space diffuser on the different extensional directions has been discussed. Furthermore, the response relationship between the linear equation and the space diffuser's hydrodynamic performance has been constructed. This study provides the foundation and scientific support for the subsequent research about the 3D stacking design of the diffuser on the pump.

## 2. Computational Domain and Meshing

### 2.1. Computational Domain

The space diffuser of the multistage submersible pump is selected as the research object (shown in Figure 2). The main design parameters are shown in Table 1. In addition, the design flow $Q = 80$ m³/h, the single pump head $H = 18$ m, and the rotation speed $n = 2800$ r/min.

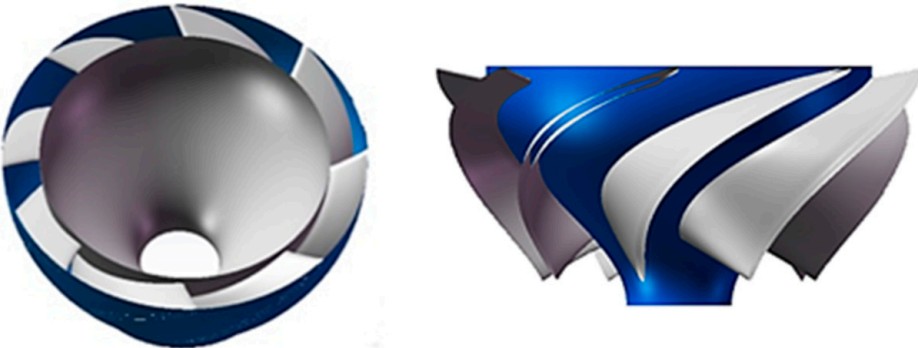

**Figure 2.** Multistage submersible pump diffuser model.

**Table 1.** The model's main parameters.

| The Impeller Parameters | | | |
| --- | --- | --- | --- |
| **Parameters** | **Values** | **Parameters** | **Values** |
| Outlet Diameter | 138.5 mm | Outlet Width | 20 mm |
| Blade Number | 7 | | |
| The Space Diffuser Parameters | | | |
| **Parameters** | **Values** | **Parameters** | **Values** |
| Inlet Diameter | 166 mm | Outlet Diameter | 92 mm |
| The Hub Wrap Angle | 83° | The Shroud Wrap Angle | 60° |
| Blade Number | 8 | | |

Because the research results of the two-stage model are basically the same as the multistage model, this paper used a two-stage model to avoid the long calculation time [27]. A 3D model of the pump was built using CFturbo in Figure 3 and, in order to improve the accuracy of calculation, the inlet and outlet pipes have been appropriately extended to allow the flow to develop fully.

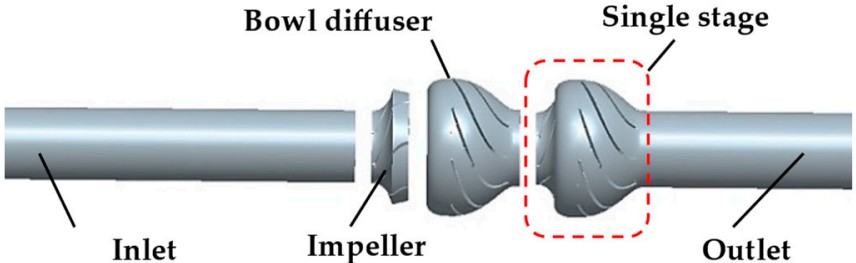

**Figure 3.** Multistage submersible pump fluid domain.

*2.2. Compute Mesh Independence Test*

ANSYS ICEM 17.0 software is used to generate the mesh of the calculational model, the unstructured mesh is selected to consider the calculation robust and the complexity structure of the space diffuser, and the local meshes have been intended. The quality of the whole mesh is above 0.3, and the pump head and diffuser efficiency are applied to verify the mesh's independence. Table 2 shows that the value of the pump head and the space diffuser efficiency tends to be stable with the mesh number increasing, when it reaches 6.8 million or more, the value is unchanged. Considering the calculation cost and accuracy, the total grid number is 6.8 million, shown as in Figure 4.

**Table 2.** The verification of the mesh independence.

| Parameter | Number of Grid $N$ ($\times 10^4$) | Head $H$/m | Diffuser Efficiency $\eta$/% |
|---|---|---|---|
| | 302 | 33.20 | 95.04 |
| | 410 | 33.04 | 95.00 |
| Value | 553 | 32.86 | 94.91 |
| | 680 | 32.74 | 94.88 |
| | 837 | 32.69 | 94.88 |
| | 988 | 32.70 | 94.88 |

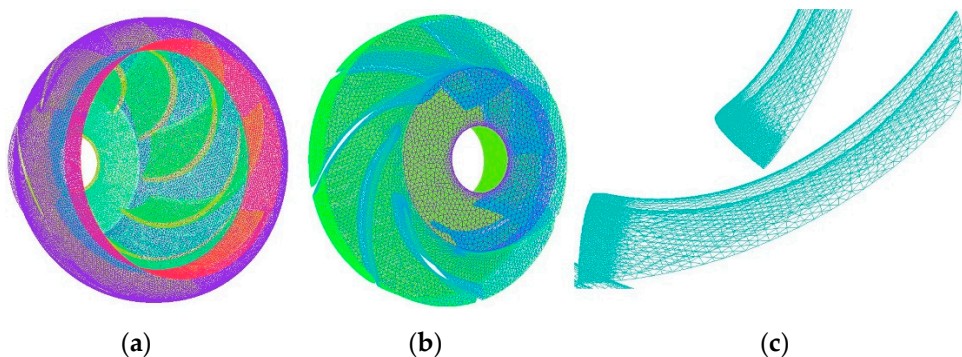

| (a) | (b) | (c) |

**Figure 4.** Mesh of the calculational model in multistage submersible pump fluid domain. (**a**) mesh of the diffuser. (**b**) mesh of the impeller. (**c**) details of the impeller blade mesh.

The CFX 17.1 software is used for numerical simulation of the whole fluid field of the multistage submersible pump. The standard RNG k-$\varepsilon$ turbulence model [28,29], scalable wall function, and no-slip boundary have been selected. The total pressure (0 Pa) is applied to the inlet boundary, and the mass flow rate (23.564 kg/s) is set as the outlet boundary. The fluid model has no heat transfer. The convergence criteria is $5 \times 10^{-5}$.

*2.3. Experiment and Simulation Validation*

The motor rotating speed of the multistage submersible pump has been affected by the load, voltage, and temperature during the actual characteristics' performance test. Therefore, the experimental data converts to the rated speed (i.e., 2800 r/min) according to the law of similarity to ensure the experimental result is comparable. In order to verify the

accuracy of the simulation method, the experiment was conducted on a professional testing platform [30–32]. And, the numerical and experimental head of the multistage submersible pump under the (0.8–1.1) $Q$ are compared in Figure 5. It can be seen that the numerical head is a good match with the experimental result. The experimental head is generally lower than the numerical result for the simplification of the 3D model. The relative deviation of the head between the experiment and number is 1.08% at the designed flow rate, which indicates that the numerical method in this paper can accurately predict the performance of the multi-stage submersible pump.

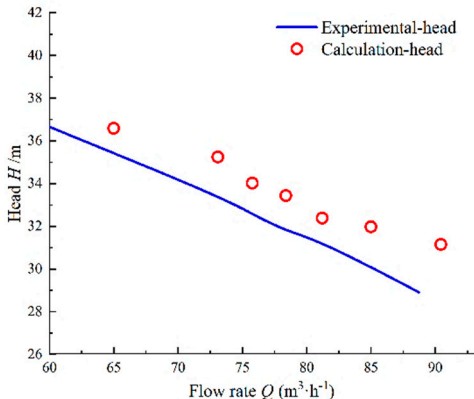

**Figure 5.** Comparison curves of pump head between the numeric and experiment.

## 3. Preliminary Design and Analysis

### 3.1. Wrap Angle Difference

The wrap angle is selected as the optional parameter for the space diffuser performance, which is defined as follows:

$$\Delta\phi = \phi_{hub} - \phi_{shroud} \tag{1}$$

where, $\phi_{hub}$ is the diffuser wrap angle of the hub, $\phi_{shroud}$ is the diffuser wrap angle of the shroud.

### 3.2. Linear Equation of the Space Diffuser Trailing Edge

To ensure the wrap angle difference of each flow surface from the hub to the shroud shows linear variation, the linear equation is applied to control the profile line of the space diffuser trailing edge. The equation represents the distribution of the space diffuser trailing edge profile line relative to the constant axial surface, which is shown as follows:

$$f(Sp) = a_0 + a_1 Sp \tag{2}$$

where $Sp$ is the extensional coefficient, $Sp \in [0, 1]$, 0 stands for the hub surface, 1 is the shroud surface; $a_0, a_1$ are optimized parameters (adjust $a_0$ and $a_1$ can get different equations). Equation (2) describes the distribution of the space diffuser on the trailing edge profile line concerning the axial surface which the wrap angle is 83°, which is the wrap angle of the original space diffuser hub. It is the difference of the wrap angle between each flow surface on the space diffuser extensional direction and the original diffuser hub. In geometry, when $Sp = 0$, $f(Sp) = a_0$ is the difference of the hub wrap angle between the new one and the original; When $Sp = 1$, $f(Sp) = a_0 + a_1$ is the difference of the shroud.

### 3.3. Evaluation Indictor

The efficiency, the static pressure energy recovery coefficient, and the non-uniform are selected to evaluate the hydrodynamic performance of the space diffuser. Where the efficiency $\eta$ denotes the efficiency of the space diffuser, not the efficiency of the multistage submersible pump. The static pressure energy recovery coefficient $C_p$ [8] indicates the ability to convert the kinetic energy into static pressure energy when the fluid flows

through the space diffuser, and the increasing $C_p$ indicates the ability of the static pressure energy recovery enhancement. The non-uniform $\zeta_i$ is a quantitative parameter of the flow uniformity at the diffuser trailing edge. The smaller non-uniform $\zeta_i$ value stands for the more uniform flow at the space diffuser trailing edge; conversely, the larger non-uniform $\zeta_i$ value is. The related equations are as follows:

$$\eta = \frac{P_{t4}}{P_{t3}} \tag{3}$$

$$C_p = \frac{P_{s4} - P_{s3}}{P_{s3}} \tag{4}$$

$$\zeta_i = \frac{1}{Q} \int_{Ai} \sqrt{(V_Z - V_{F,av,i})^2} \, dA \tag{5}$$

where $P_{t3}$ is the inlet total pressure of the space diffuser (Pa), $P_{t4}$ is the outlet total pressure of the space diffuser (Pa), $P_{s3}$ is the inlet static pressure of the space diffuser (Pa), $P_{s4}$ is the outlet static pressure of the space diffuser (Pa), $Q$ is the design flow of the multistage submersible pump (m$^3$/h), $V_Z$ is the local axial velocity of the diffuser outlet surface (m/s), and $V_{F,av,i}$ is the average velocity of the space diffuser outlet surface (m/s).

### 3.4. Initial Value of the Wrap Angle Difference

Linear Equation (2) controls the profile line of the space diffuser trailing edge by adjusting the optimized parameters $a_0$ and $a_1$. Keeping the $\phi_{hub}$ is 83° and changing the shroud wrap angle, seven different wrap angle difference cases have been analyzed by the CFD method. The relationship between the profile line of the space diffuser trailing edge and the linear equation is shown in Figure 6, which includes the cases of the wrap angle difference $\Delta\phi$ is $-20°$, $-15°$, $-10°$, $0°$, $10°$, $20°$ and $23°$. To make it clear, the horizontal coordinate of the rectangular coordinate system is the wrap angle difference, and the results are presented in Figure 7.

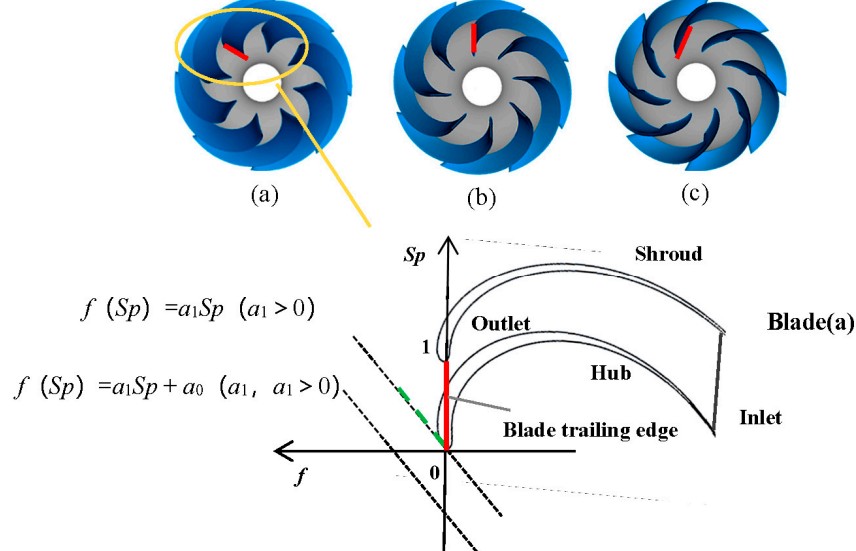

**Figure 6.** The relationship between the trailing edge of the blade and the linear equation. (**a**) $\Delta\phi < 0°$, $a_1 > 0$; (**b**) $\Delta\phi = 0°$, $a_1 = 0$; (**c**) $\Delta\phi > 0°$, $a_1 < 0$.

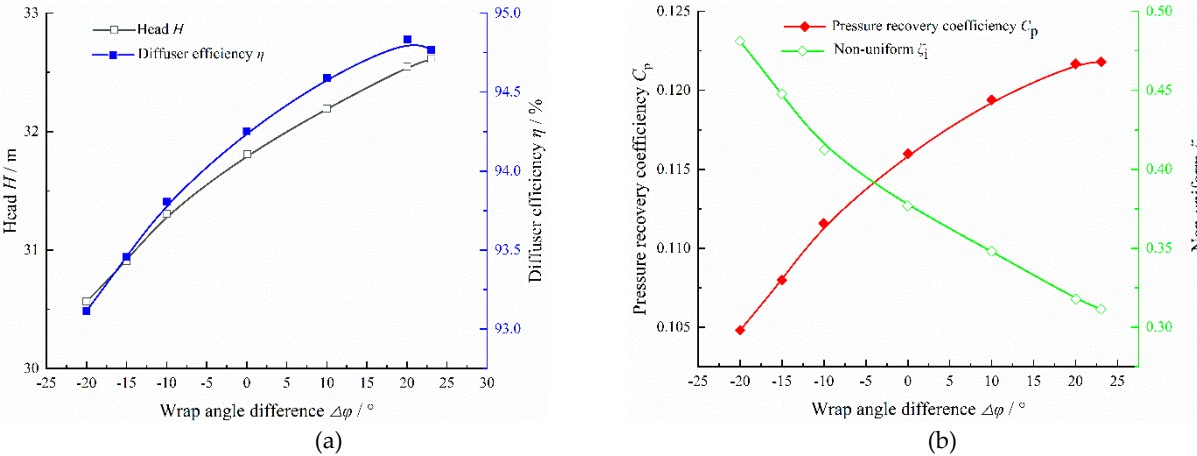

(a)

(b)

**Figure 7.** The performance of the space diffuser with the different wrap angle difference $\Delta\phi$. (**a**) The pump head and space diffuser efficiency (**b**) The pressure recovery coefficient and the non-uniform.

When $a_1 > 0$ (where $a_1$ represents the optimized parameter), the wrap angle difference is less than zero, indicating that the wrap angle of the hub is smaller than that of the shroud, as shown in Figure 6a. In Figure 7, it can be observed that the head, the static pressure energy recovery coefficient $C_p$, and outlet non-uniform $\zeta_i$ of the multistage submersible pump are comparatively lower when the wrap angle difference is less than zero degrees (i.e., $\Delta\phi < 0°$). This indicates that the performance of the multistage submersible pump is poorer. Figure 8 illustrates the streamlined distribution on the hub blade to blade surface under the different wrap angle difference conditions. A large amount of the secondary flow separation vortices appears on the corner region of the hub suction surface at the blade trailing edge in Figure 8, which occupies approximately 1/3 of the blade-to-blade flow passage.

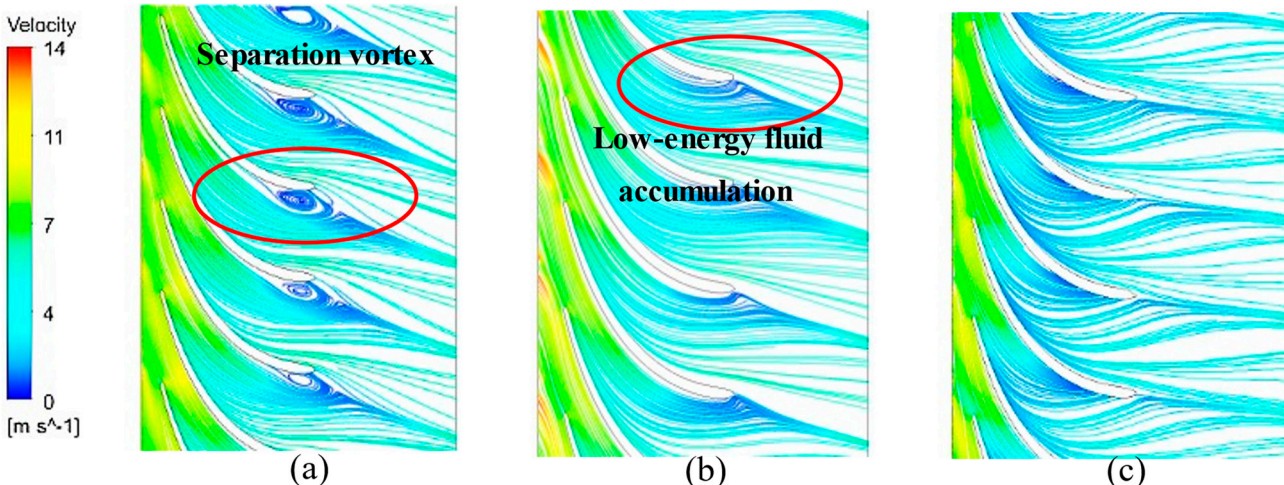

(a)          (b)          (c)

**Figure 8.** Streamlines on the hub surface. (**a**) $\Delta\phi < 0°$ (**b**) $\Delta\phi = 0°$ (**c**) $\Delta\phi > 0°$.

The mechanisms of the secondary flow separation vortices on the corner region include: (a) The shape and the difference in curvature of the space diffuser inlet flow passage causes the total pressure on the shroud surface to be higher than that on the hub surface. The pressure difference between the shroud and hub surface causes the extensional pressure difference, which is shown in Figure 9 L region. The low-energy fluid on the suction surface is driven by the extensional pressure difference to be swept from the shroud to the hub, and it mixes with the mainstream from the space diffuser inlet. The low-energy fluid overcomes the adverse pressure gradient and leaves the suction surface to flow towards the pressure

surface. (b) In the middle of the space diffuser, the kinetic energy of the fluid decreases due to the work done by the diffuser. Under the action of adverse pressure gradient along the mainstream flow direction, low-energy fluid gradually accumulates, is driven by the transversal pressure difference, leaves off the pressure surface, and flows towards the suction surface corner region. Upon reaching the suction surface, it is unable to overcome the extensional pressure difference from the shroud to the hub and is forced to flow back to the inlet. Then, it interacts with the low-energy fluid flowing from the suction surface to the pressure surface of the diffuser inlet region (generating in (a) section), forming a secondary flow corner region separation vortex.

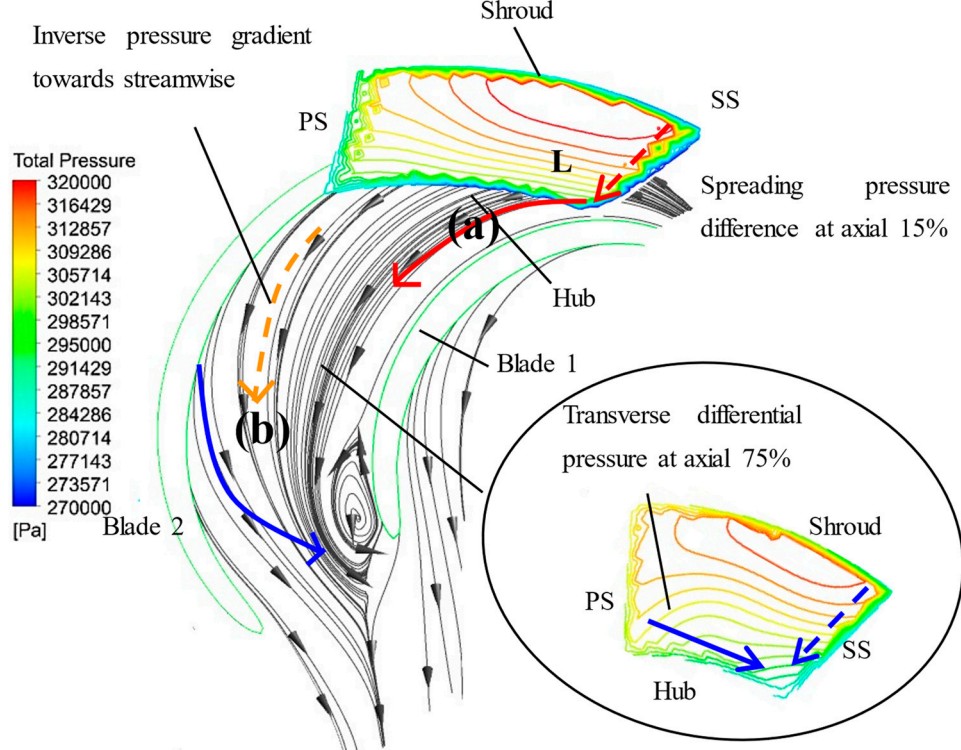

**Figure 9.** Limiting streamlines of hub surface and total pressure contour of the diffuser ($\Delta\phi = 0°$).

In Figure 8b, adjusting the parameter $a_1 > 0$ and keeping $\Delta\phi = 0°$, the separation vortex at the hub trailing edge corner region weakens, however, there is still a significant accumulation of the low-energy fluid in the corner region. In Figure 8c, adjusting the parameter $a_1 < 0$ and keeping $\Delta\phi > 0°$, the separation vortex at the hub trailing edge corner region almost disappears, and there is no significant accumulation of the low-energy fluid. Based on the above observations, it can be concluded that when the diffuser wrap angle difference is negative, the performance of the diffuser is poorer. Therefore, it is recommended to select a positive wrap angle difference of the space diffuser ($a_1 \leq 0$, $\Delta\phi \geq 0°$) for further research.

## 4. Results and Discussion

### 4.1. Optimization Cases in Positive Wrap Angle Difference

To investigate the effects of the variable wrap angle difference within the range of (0, 35°) on the internal flow mechanism of a space diffuser, two optimization schemes were designed based on the positive wrap angle difference ($a_1 < 0$ and $\Delta\phi \geq 0°$). For the original space diffuser model, the hub wrap angle $\phi_{hub}$ is 83°, and the shroud wrap angle $\phi_{shroud}$ equals to 60°, so the linear equation for the space diffuser trailing edge profile line is $f(Sp) = -23Sp$. Two optimization schemes are as follows:

Scheme A: $a_0 = 0$, $a_1 \in [-35, 0]$, the corresponding linear equation is $f(Sp) = a_1 Sp$, $a_1 = -\Delta\phi$ and is a line passing through the origin. The hub wrap angle remains unchanged

at 83°, while the shroud wrap angle decreases from 83° to 48°. Scheme B: $a_1 \in [-35, 0]$, the related linear equation doesn't pass through the origin, $f(Sp) = a_1 Sp + a_0$, $a_0 + a_1 = -23$, $a_1 = -\Delta\phi$, $a_0 = \phi_{hub2} - \phi_{hub1}$, where $\phi_{hub1}$ and $\phi_{hub2}$ are the hub wrap angle before and after optimization. The shroud wrap angle stays the same at 60°, while the hub wrap angle increases from 60° to 83°.

### 4.2. Analysis of the Optimization Results

Figure 10 is about the relationship between the wrap angle difference with the space diffuser performance. In Figure 10, the performance of the multistage submersible pump can be improved by increasing the wrap angle difference within the range of [0, 35°]. However, two optimization schemes, one increasing the hub wrap angle and the other decreasing the shroud wrap angle, have different effects on the improvement of pump performance. Scheme A increases the wrap angle difference by decreasing the shroud wrap angle, and then the space diffuser hydraulic performance (such as the diffuser efficiency $\eta$, head $H$, non-uniform $\zeta_i$, and static pressure energy recovery coefficient $C_p$) unidirectional improves with the increase of the wrap angle difference. Nevertheless, the hydraulic performance of the space diffuser in scheme B, which increases the wrap angle difference by increasing the hub wrap angle, exhibits a characteristic of initially increasing and then decreasing, a maximum value is at $\Delta\phi = 20°$.

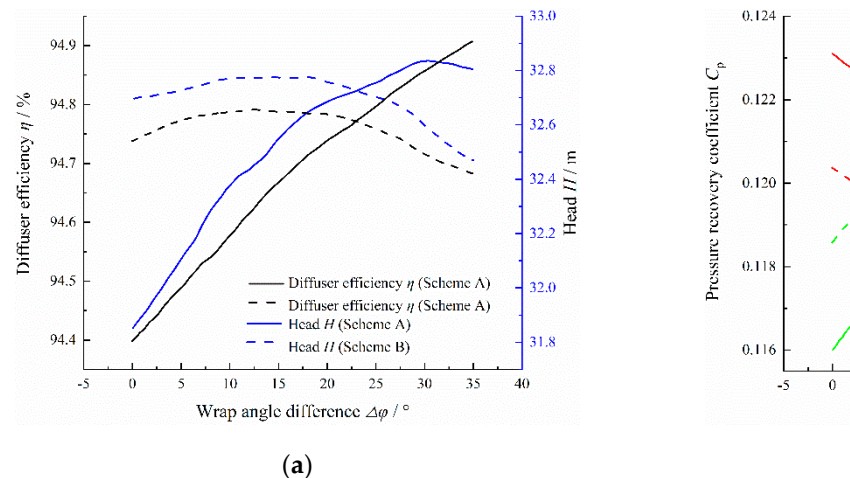
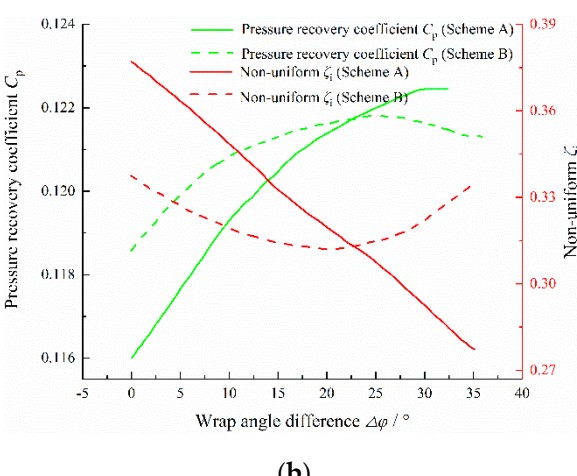

| (a) | (b) |

**Figure 10.** The relation between $\Delta\phi$ and performance of diffuser. (**a**) The head and diffuser efficiency (**b**) The pressure recovery coefficient and non-uniform.

### 4.3. Hydrodynamic Analysis of the Variable Shroud Wrap Angle (Scheme A)

To investigate the mechanism of the decreasing shroud wrap angle to improve the hydraulic performance of the space diffuser, the inner flow field of the space diffuser with the wrap angle difference of $\Delta\phi = 0°(f(Sp) = 0)$ and $\Delta\phi = 30°$ $(f(Sp) = -30Sp)$ was analyzed. Figure 11 shows the corresponding trailing edge profiles of the space diffuser. From Figure 11, it can be observed that the wrap angle decreases continuously from the hub to the shroud on each flow surface. Compared to the wrap angle difference of 0°, the circumferential inclination at the blade trailing edge for the 30° wrap angle difference case increases.

Figure 12 shows the static pressure and streamlines on the space diffuser outlet. The static pressure at the diffuser outlet significantly increases when the wrap angle difference equals to 30° ($\phi_{shroud} = 53°$). However, the static pressure recovery coefficient increases by 0.7% compared to the case of the 0° ($\phi_{shroud} = 83°$). The higher static pressure energy recovery coefficient causes the swirl velocity to decrease, and the outlet flow becomes more uniform. In this case ($\Delta\phi = 30°$, $\phi_{shroud} = 53°$), the curvature radius of each flow surface decreases by reducing the shroud wrap angle. Then the blade attack angle increases, and the workability of the space diffuser is enhanced. When the fluid flows through the space

diffuser blade, it can convert more rotational kinetic energy into static pressure energy, thereby causing the static pressure recovery coefficient to increase. In Figure 13, the static pressure loading curve in the case $\Delta\phi = 30°, \phi_{shroud} = 53°$ is obviously larger than the $\Delta\phi = 0°, \phi_{shroud} = 83°$ case. Therefore, as the wrap angle difference increases (i.e., the shroud wrap angle decreases), the static pressure energy recovery coefficient increases.

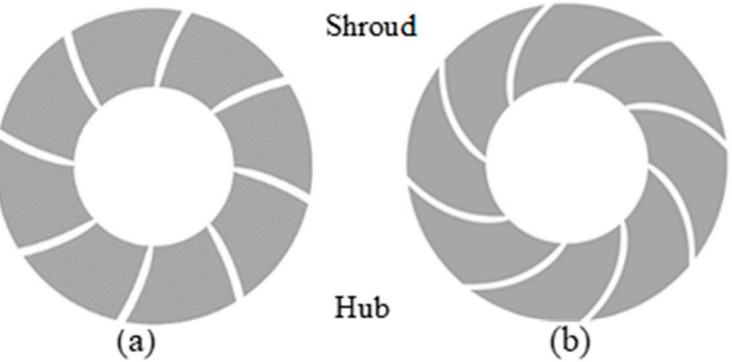

**Figure 11.** Diffuser blade trailing edge profile. (**a**) $\Delta\phi = 0°$ (**b**) $\Delta\phi = 30°$.

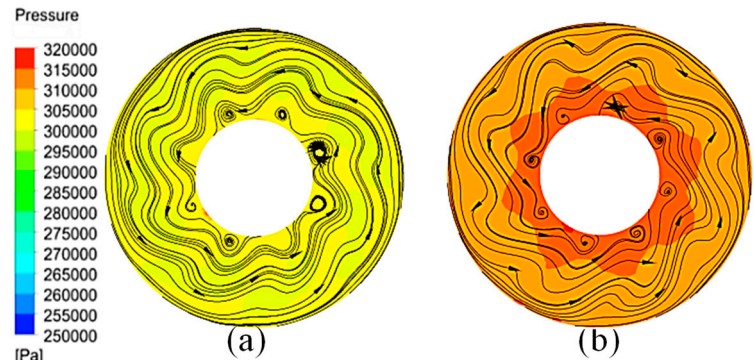

**Figure 12.** Comparison of static pressure and streamlines of the diffuser outlet. (**a**) $\Delta\phi = 0°$ (**b**) $\Delta\phi = 30°$.

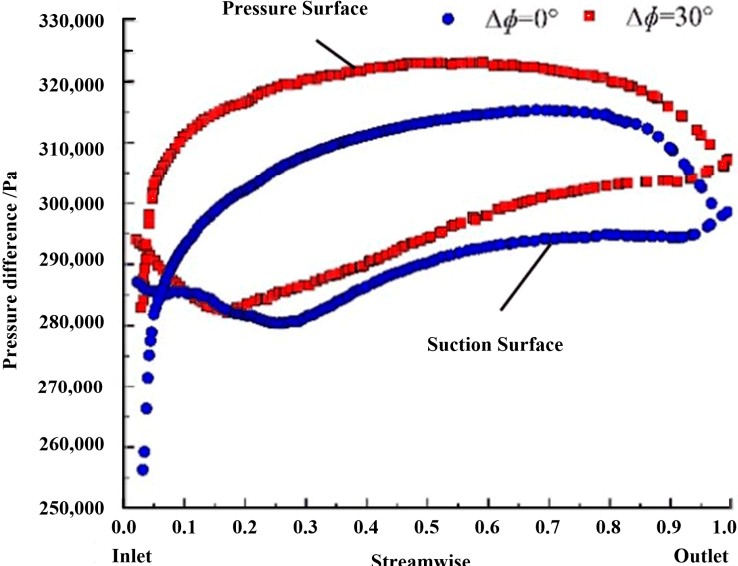

**Figure 13.** Static pressure comparison on the shroud surface of the diffuser.

To explain the reason for the lower uniformity and the hydraulic efficiency when the wrap angle difference is $0°$, the distribution of the total pressure, velocity, and static pressure

has been discussed. A large amount of the low-energy fluid accumulated in the blade trailing edge corner region in the hub suction surface, driven by the transversal pressure difference from the shroud to the hub and the extensional pressure difference from the pressure surface to the suction surface, when the wrap angle difference is 0°, in Figure 14a,b. This phenomenon has also been confirmed in Figure 15a. When the wrap angle difference is 0°, the working fluid in the space diffuser gradually decreases under the influence of the transversal adverse pressure gradient, leaves off the shroud surface, and flows to the hub surface by the action of the extensional pressure difference from the shroud to the hub. Abundant low-energy fluid appears and collects on the blade trailing edge, deteriorating the uniformity of the outlet flow. In Figure 16a,b, the low-energy fluid is generated at the trailing edge of both the shroud and the hub. There is a significant difference in mainstream velocities. They gathered at the space diffuser outlet, causing severe energy loss, resulting in the poor uniformity of the diffuser outlet and lower hydraulic efficiency.

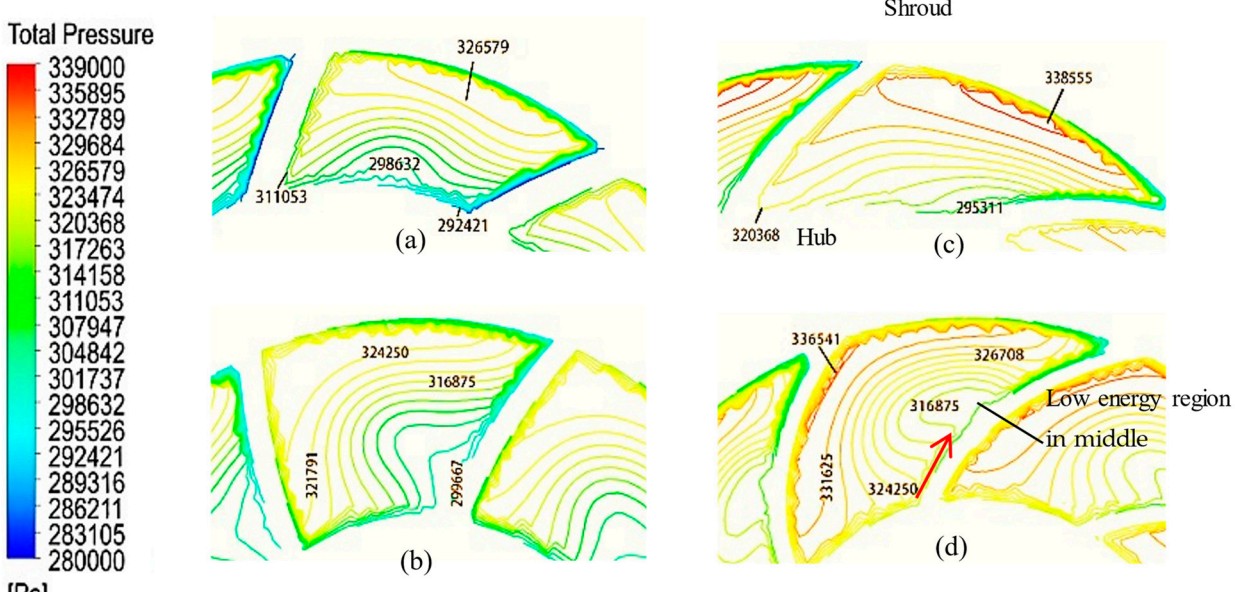

**Figure 14.** Total pressure contour of the diffuser. (**a**) Section at axial 15%. $\Delta\phi = 0°$ (**c**) Section at axial 15% $\Delta\phi = 30°$. (**b**) Section at axial 90% $\Delta\phi = 0°$ (**d**) Section at axial 90% $\Delta\phi = 30°$

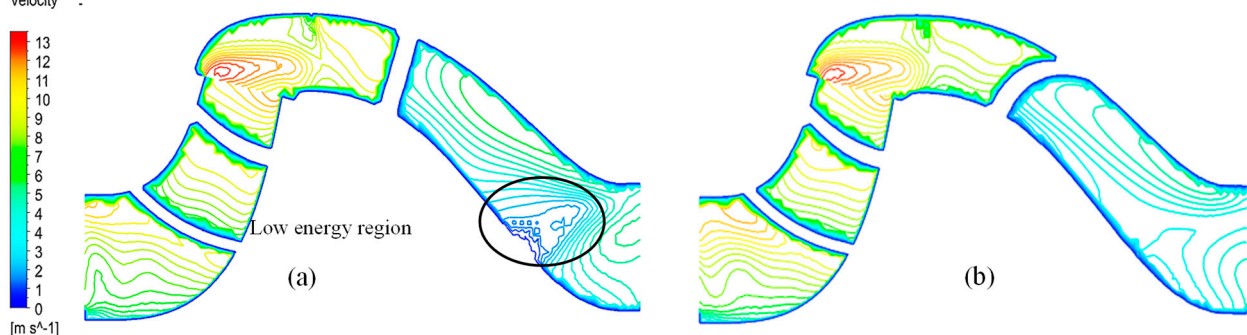

**Figure 15.** Velocity distribution on the meridional plane of the diffuser. (**a**) $\Delta\phi = 0°$ (**b**) $\Delta\phi = 30°$.

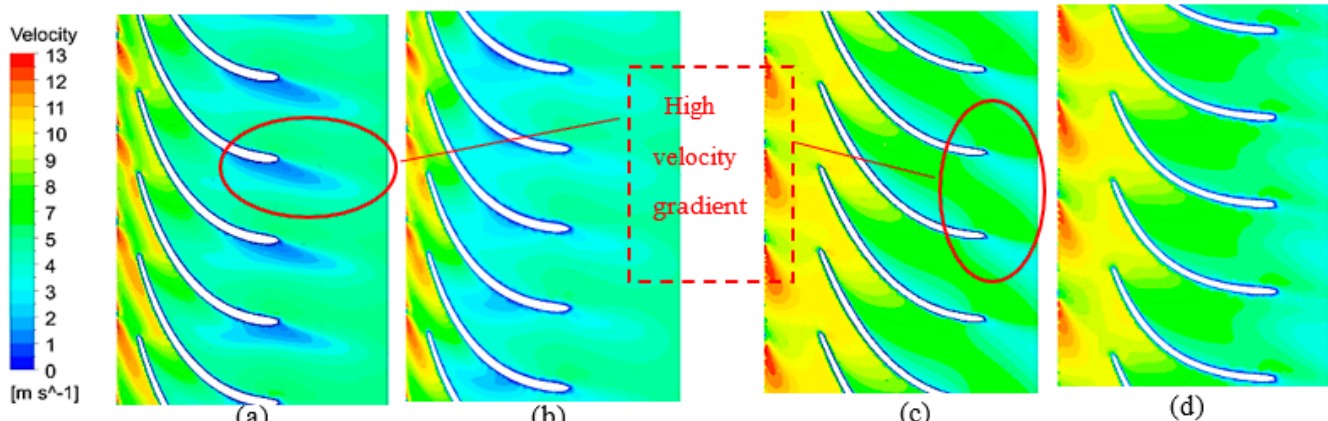

**Figure 16.** Contrast the stream surface of the diffuser wake region: (**a**) $\Delta\phi = 0°$ (Hub) (**b**) $\Delta\phi = 30°$ (Hub) (**c**) $\Delta\phi = 0°$ (Shroud) (**b**) $\Delta\phi = 30°$ (Shroud).

In Figures 17 and 18, a quantitative analysis of the extensional and transversal pressure difference can be conducted to further explain the reason for the improvement in the uniformity and the hydraulic efficiency of the diffuser outlet. Figure 17 illustrates the extensional pressure difference between the shroud and the hub. The extensional pressure difference at the leading edge is lower for the higher wrap angle difference ($\Delta\phi = 30°$) compared to the lower wrap angle difference ($\Delta\phi = 0°$). The increased wrap angle difference reduces the strength of the extensional secondary flow, thereby suppressing the sweep from the blade passage region between the shroud and the hub, which has been mentioned in Section 3.4. As a result, the transverse secondary flow from the suction to the pressure surface, induced by the blade inlet extensional secondary flow, is weakened. Meanwhile, as shown in Figure 18, the transversal pressure difference from the inlet to point A of the space diffuser gradually increases, increasing the transversal secondary flow from the pressure surface to the suction surface. The transversal secondary flow counteracts the fluid flowing from the suction surface to the pressure surface, thereby suppressing the generation of the corner region separation vortices.

As the shroud wrap angle reduces, the curvature radius of the space diffuser blade decreases, and the main working region of the blade moves forward. Therefore, in Figure 18, the transversal pressure difference from the pressure surface to the suction surface in the BC section reduces, and the transversal secondary flow weakens, so the low-energy fluid at the corner region of the hub suction surface in Figure 15a decreases. Meanwhile, in Figure 17, the extensional pressure difference from the shroud to the hub reverses to that from the hub to the shroud, the original low-energy fluid, caused by the backflow fluid, has been drained to the lower pressure region in Figure 14d. The separation vortex, located on the corner region, has been inhibited. The uniformity of the space diffuser outflow has been enhanced; the non-uniformity value reduced by 9%. The low-energy fluid in the trailing edge region has been suppressed in Figure 16b,d. The velocity difference between the low-energy fluid and the mainstream decreases, which reduces the mixing losses at the diffuser outlet, resulting in a 0.48% improvement in the space diffuser hydraulic efficiency.

The whole hydraulic efficiency of the second space diffuser and the impeller has been compared under two scheme conditions in Figure 19. The second hydraulic efficiency unidirectional increases with the wrap angle difference increasing. The larger wrap angle difference, which is caused by decreasing the shroud wrap angle, can effectively suppress the low-energy fluid accumulation. The separation vortex in the hub surface corner region has been inhibited, and it improves the hydraulic performance of the space diffuser. When $\Delta\phi > 30°$, the overall efficiency increases slowly and the manufacture becomes difficult, the optimal selection is that the wrap angle difference equals $30°$. When the equation $f(Sp) = a_0 + a_1 Sp$ passes through the origin, the parameter $a_1$ is about $-30$, the $\phi_{hub} = 83°$, $\phi_{shroud} = 53°$; And the space diffuser efficiency ($\eta$) is 94.9%, the second diffuser and the

impeller efficiency ($\eta_t$) is 79.6%, the static pressure recovery coefficient ($C_p$) is 0.123, the non-uniformity ($\zeta_i$) is 0.290.

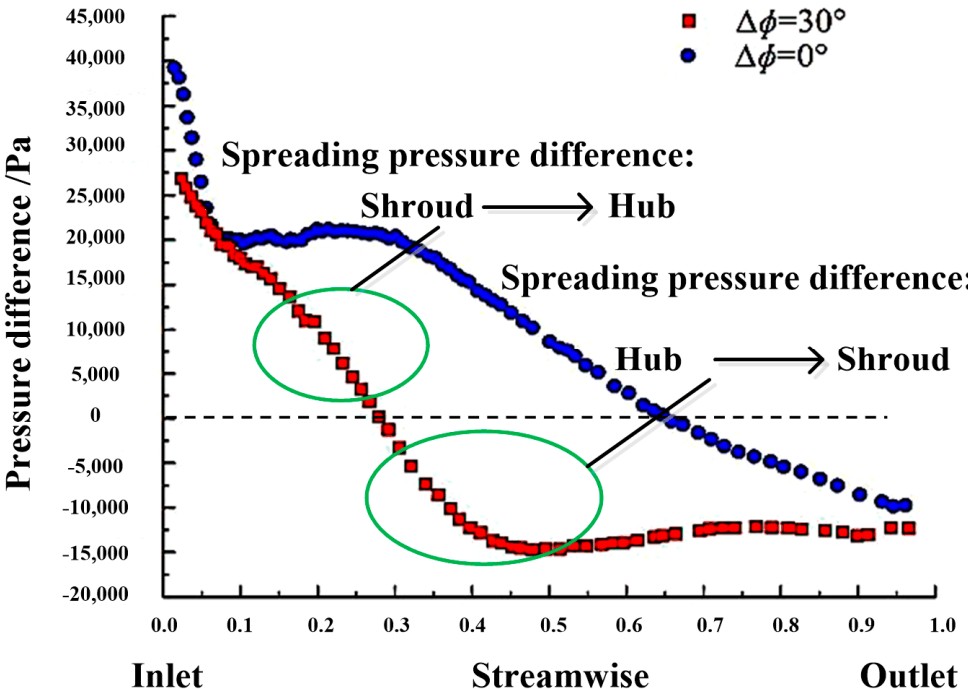

**Figure 17.** Comparison of extensional pressure difference from shroud to hub on the suction surface of the diffuser.

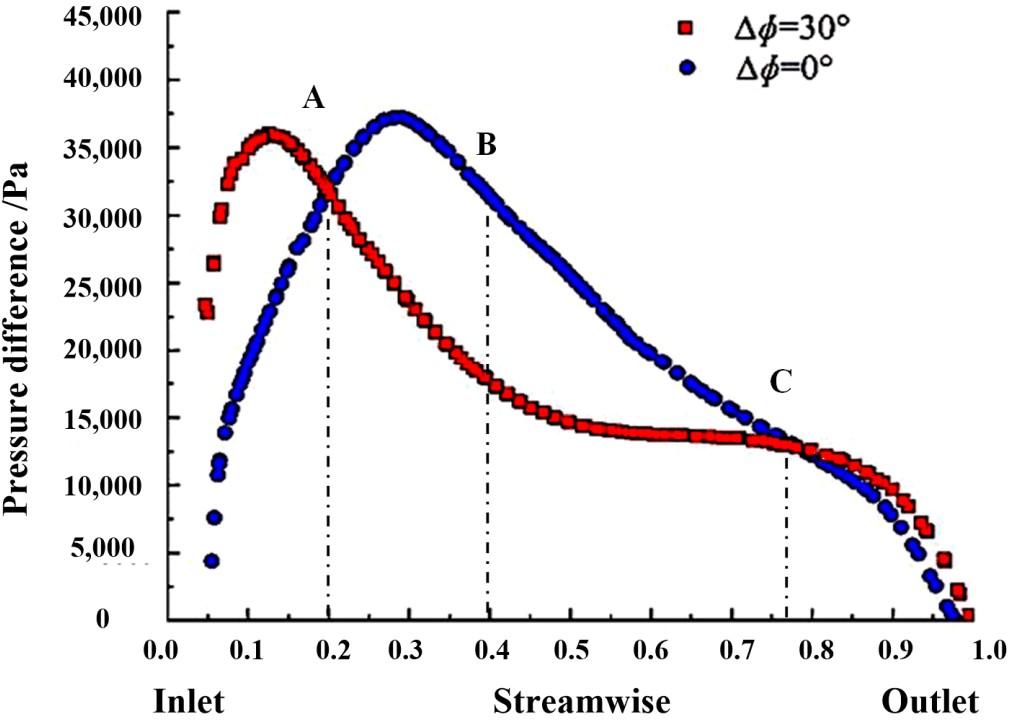

**Figure 18.** Comparison of transversal pressure difference from pressure surface to suction surface on hub surface of the diffuser.

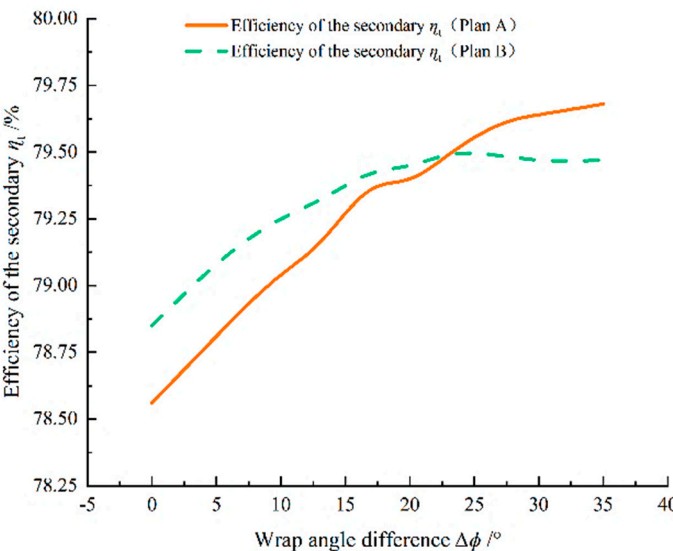

**Figure 19.** The relation between warp angle difference $\Delta\phi$ and the secondary efficiency.

### 4.4. Hydrodynamic Analysis of the Variable Hub Wrap Angle (Scheme B)

The wrap angle difference is varied by adjusting the hub wrap angle, the shroud wrap angle keeps constant in Scheme B. The extreme value of the hydraulic efficiency $\eta$, uniformity $\zeta_i$, and the static pressure energy recovery coefficient $C_p$ of the space diffuser appears at the wrap angle difference equals $20°$ ($\Delta\phi = 20°$), with the wrap angle difference increasing in Figure 10. To distinguish the reason for the effect of the hub wrap angle on its hydraulic performance, three cases (such as $\Delta\phi = 0°$ ($f(Sp) = 0$), $\Delta\phi = 20°$ ($f(Sp) = -20Sp - 3$), and $\Delta\phi = 35°$ ($f(Sp) = -35Sp + 12$)) have been analyzed. In Figure 20, they are defined as "case $\Delta\phi = 0°$", "case $\Delta\phi = 20°$", and "case $\Delta\phi = 35°$" respectively.

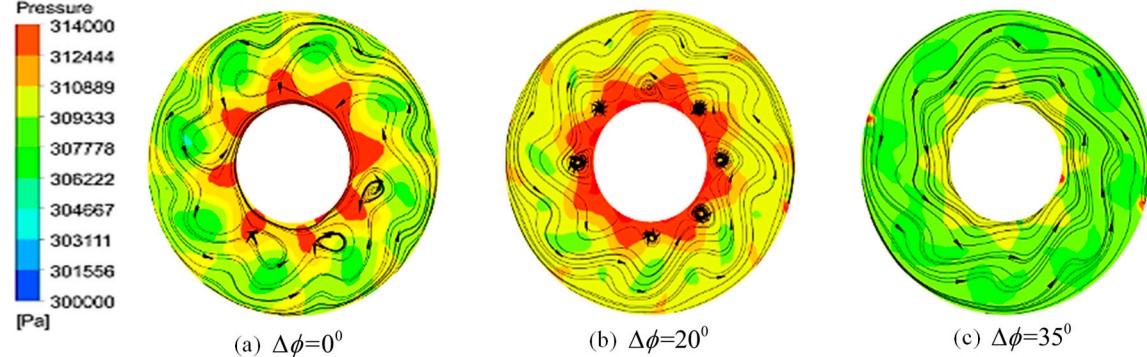

(a) $\Delta\phi=0^0$          (b) $\Delta\phi=20^0$          (c) $\Delta\phi=35^0$

**Figure 20.** Comparison of static pressure and streamlines of diffuser outlet surface under different hub wrap angles.

As shown in Figure 20b, comparing to the cases of the $\Delta\phi = 0°$ and $\Delta\phi = 35°$, the highest static pressure of the diffuser outlet surface appears at the case $\Delta\phi = 20°$ (the hub wrap angle is $80°$), and the static pressure energy recovery coefficient $C_p$ is 0.121. The results are in alignment with Figure 10. As the hub wrap angle increases, the blade curvature radius on each flow surface also increases, while the blade attack angle decreases, and the working capacity diminishes. Consequently, the adverse pressure gradient diminishes, and the reduction in mainstream fluid velocity is not obvious, thus the low-velocity fluid on the space diffuser can be controlled. As shown in Figures 21 and 22, increasing the hub wrap angle, the extensional pressure difference on the middle section of the space diffuser from the hub to the shroud increases. The low-energy fluid has been pushed to the lower pressure region on the blade suction surface middle section. The accumulation of low-energy fluids on the hub suction surface corner region has been released (in Figure 23).

The separation vortex on the corner region has been inhibited. Meanwhile, increasing the hub wrap angle, the working capacity is eroded, the transversal pressure difference on the blade middle section is reduced, the strength of the secondary flow is suppressed, and the accumulation of low-energy fluids on the hub suction surface corner region is further alleviated, thus improving the uniformity and the hydraulic efficiency of the space diffuser.

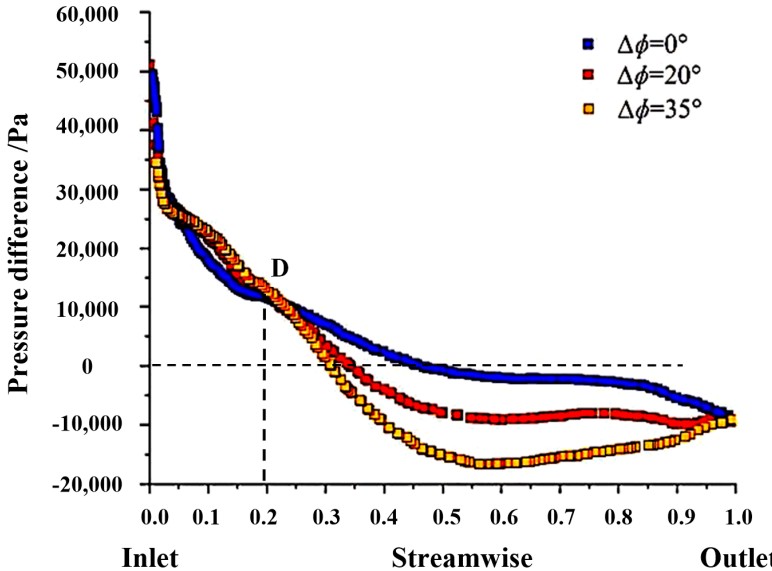

**Figure 21.** Comparison of pressure difference from shroud to hub on the suction surface.

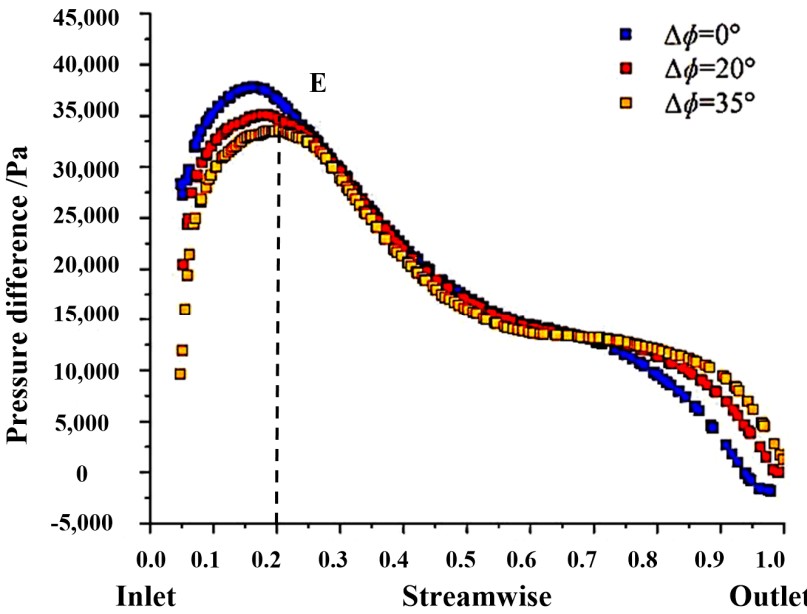

**Figure 22.** Comparison of pressure difference from pressure surface to suction surface on hub surface.

Unlike scheme A, the main load region of the space diffuser does not migrate toward the inlet as the wrap angle difference increases. It leads to an increase in the extensional pressure difference from the shroud to the hub at the space diffuser inlet, intensifying the washing from the shroud to the hub corner region in Section 3.4. Consequently, the transversal secondary flow from the suction to the pressure surface, originating from the extensional secondary flow of the space diffuser inlet, is enhanced, promoting the formation of the corner separation vortex. Simultaneously, the transversal pressure difference from the pressure to the suction surface at the diffuser inlet decreases in Figure 22. It can't entirely counteract the transversal secondary flow from the suction to the pressure surface caused

by the extensional secondary flow at the diffuser inlet. It further accelerates the generation of the separation vortex, ultimately exacerbating the diffuser's hydraulic performance.

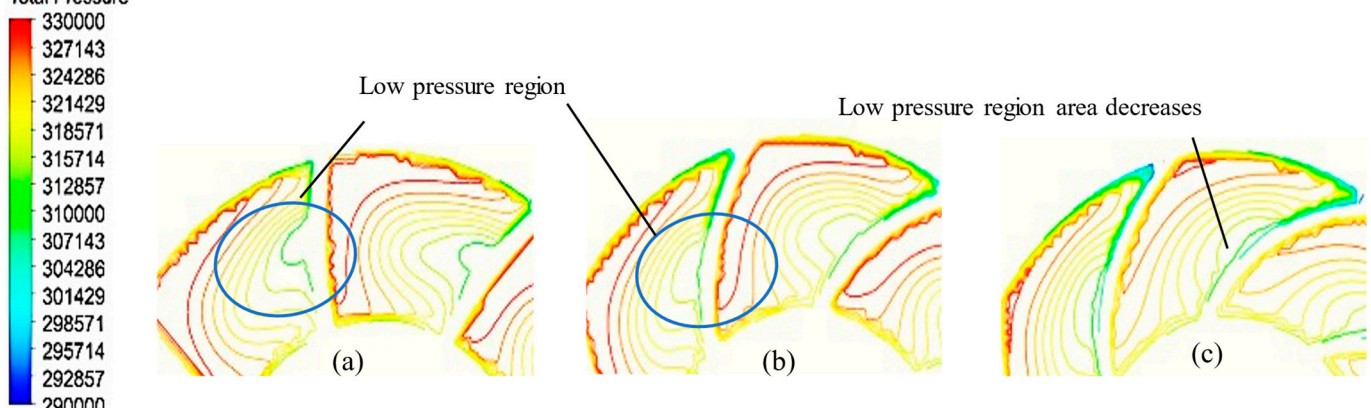

**Figure 23.** Total pressure contour of the diffuser. (**a**) Section at axial 90% $\Delta\phi = 0°$ (**b**) Section at axial 90% $\Delta\phi = 20°$ (**c**) Section at axial 90% $\Delta\phi = 35°$.

From the inlet to the E point in Figure 22, with the wrap angle difference increasing ($\Delta\phi$), the transverse secondary flow from the pressure surface to the suction surface at the diffuser inlet is weakened. Then the transverse secondary flow from the suction surface to the pressure surface caused by the inlet extensional secondary flow is strengthened, and low-energy fluid on the hub surface accumulates. Meanwhile, the extensional pressure difference on the blade middle is enhanced, and the low-energy fluid has been pushed from the hub to the shroud. However, the reducing static pressure on the shroud surface is not enough to drive the low-energy fluid to the blade's extensional middle section. So, the area of the region of the low-pressure shrikes is shown in Figure 23c.

In summary, the interaction between the increasing secondary flow at the diffuser inlet and the reducing accumulation of low-energy fluid at the diffuser middle section results in extreme values in the diffuser's hydraulic performance. When the hub wrap angle is 80° (corresponding to an incidence angle difference of 20°), the optimal hydraulic performance is achieved: the non-uniformity $\zeta_i$ is 0.310, the static pressure recovery coefficient $C_p$ is 0.122, and the diffuser efficiency $\eta$ is 94.84%.

## 5. Conclusions

A linear equation is established to regulate the trailing edge profile of the space diffuser, optimizing the wrap angle on each flow surface from the shroud to the hub in the multi-stage submersible pump, and changing the wrap angle difference. The synergistic relationship between the space diffuser trailing edge stacking configuration and the internal flow field has been discussed. Several conclusions have been obtained as follows:

(1)   The increasing wrap angle difference within a specific range can alleviate the accumulation of the low-energy fluid and suppress the generation of the secondary flow separation vortex at the hub surface. When the wrap angle difference is a positive value, the wrap angle of the space diffuser decreases gradually from the hub to the shroud surface, and the shroud wrap angle is lower than the hub. The inner secondary flow on the hub surface has been suppressed. Then the separation vortex in the corner region of the diffuser middle section has been diminished. So, the hydraulic efficiency with the positive wrap angle difference exceeds the negative wrap angle difference. That is, the space diffuser hydraulic performance with the negative stacking style ($\Delta\phi > 0°, a_1 < 0$) of the trailing edge is better than the positive one ($\Delta\phi < 0°, a_1 > 0$).

(2)   Secondary flow scouring induced by the extensional or transversal pressure differences of the diffuser leading edge, coupled with the obstructive effect of the exten-

sional or transversal pressure differences on the diffuser trailing edge on low-energy fluid, ultimately triggers the generation of the hub-corner separation vortex.

(3) To investigate the effects of the variable shroud and hub wrap angles on the internal flow field and hydraulic performance of the space diffuser, two schemes have been conducted, by reducing the impeller wrap angle and increasing the hub wrap angle within the range of $[0, 35°]$ to enhance the wrap angle difference. When the hub wrap angle keeps constant and the shroud wrap angle reduces, the transverse secondary flow from the pressure surface to the suction surface at the middle section of the space diffuser is weakened, and the accumulation of low-energy fluid at the corner region is suppressed. Meanwhile, the extensional secondary flow from the hub to the shroud surface is enhanced, which drains the low-energy flow to the middle of the blade and flows out smoothly. Therefore, in the negative stacking style of the trailing edge scheme ($\Delta\phi < 0°, a_1 > 0$) with a reduced shroud wrap angle, the secondary flow scouring at the leading edge is suppressed, and the low-energy fluid at the trailing edge is drained, both of which contribute to the overall improvement in the space diffuser's hydraulic performance, and the optimum case at $\Delta\phi = 30°, f(Sp) = -30Sp$.

(4) When the shroud wrap angle remains constant and the hub wrap angle increases, the secondary flow from the hub to the shroud at the middle region of the space diffuser increases, which leads to the number of the accumulation of low-energy fluid at the corner region being reduced. However, the intensification of extensional scouring from the shroud to the hub at the inlet and the secondary flow from the suction to the pressure surface to appear. It promotes the formation of separation vortexes. As a result, in the negative stacking style of the trailing edge scheme ($\Delta\phi > 0°, a_1 < 0$) with increased hub wrap angle, the secondary flow scouring at the leading edge is intensified, and the low-energy fluid at the trailing edge is drained, leading to a mutual cancellation effect that results in the appearance of performance extremes, and the optimum case at $\Delta\phi = 20°, f(Sp) = -20Sp - 3$.

**Author Contributions:** Conceptualization, methodology, and formal analysis, H.Z.; formal analysis, visualization, supervision, project administration, P.C.; visualization, supervision, project administration, D.N.; writing—original draft preparation and data curation, X.G.; writing—review and editing, B.H.; writing—review and editing, R.Z. All authors have read and agreed to the published version of the manuscript.

**Funding:** This research was funded by the National Natural Science Foundation of China (No.52376024).

**Data Availability Statement:** Data are contained within the article.

**Conflicts of Interest:** Author Bo He was employed by the company Shanxi Wofeng Fluid Technology Co., Ltd., the remaining authors declare that the research was conducted in the absence of any commercial or financial relationships that could be construed as a potential conflict of interest.

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
