# Peer review of "Investigation on the Impact of Trailing Edge Stacking Styles on Hydraulic Performance in the Multistage Submersible Pump Space Diffuser"

_processes, doi:10.3390/pr11123358_

Round 1

Reviewer 1 Report

Comments and Suggestions for Authors

The manuscript presents numerical investigation on the relationship of trailing edge and the performance of the multistage submersible pump space diffuser. A linear equation is established, and it is useful for the design of the multistage pupm space diffuser. I suggest that the manuscript can be accepted if my following concers are addressed:

1. It seems to me that the mechanisms of the hydraulic loss are absent in the introduction. I suggest that the authors should include why the vortex initialize in the introduction and the difficulties to study the vortex in both experiments and simulations.

2. Please add reference to line 94.

3. It would be nice to include the mesh figure of the problem in section 2.2.

4. In section 2.3, where do the experiment results come from. Please add reference (from literature) or statements of the experiments (from author's team).

5. I suggest label the two y-axises in different colors in Figure 6 and 7, which should be consistent with the color in the figure.

Author Response

  1. It seems to me that the mechanisms of the hydraulic loss are absent in the introduction. I suggest that the authors should include why the vortex initialize in the introduction and the difficulties to study the vortex in both experiments and simulations.

Response: Thanks for the reviewer’s valuable recommendations. The research on the hydraulic loss has been added in lines 48-53.

  1. Please add reference to line 94.

Response: Thanks for the reviewer’s valuable suggestions. The reference has been added in line 100.

  1. It would be nice to include the mesh figure of the problem in section 2.2

Response: Thanks for the reviewer’s valuable suggestions. The mesh of impeller and diffuser had been added in Figure 4.

  1. In section 2.3, where do the experiment results come from. Please add reference (from literature) or statements of the experiments (from author's team)

Response: Thanks for the reviewer’s valuable suggestions. Three references had been added in the line 133-134.  

  1. I suggest label the two y-axises in different colors in Figure 6 and 7, which should be consistent with the color in the figure.

Response: Thanks for the reviewer’s valuable suggestions. The modified had been conducted in Figure 7 and Figure 10, which label the two y-axises in different colors. The revision section shown in lines 186 and 260.

Reviewer 2 Report

Comments and Suggestions for Authors

This manuscript focus on the hydraulic performace in the multistae submersible pumple space diffuser by the study on the trailing edge stacking sytels and the  result accurately presents the relationships. The manuscript is worthy to be published in Processes. However, there are still tiny problems for this paper like: Please improve the resolution of the figures, specilly the figures  for the lines.

Author Response

This manuscript focus on the hydraulic performace in the multistae submersible pumple space diffuser by the study on the trailing edge stacking sytels and the  result accurately presents the relationships. The manuscript is worthy to be published in Processes. However, there are still tiny problems for this paper like: Please improve the resolution of the figures, specilly the figures for the lines..

Response: Thanks for the reviewer’s valuable suggestions. Several resolution of the figures have been improved, such as the Figure 7 and 10. The modified section is shown in Figure 7 and 10.

Reviewer 3 Report

Comments and Suggestions for Authors

Review report of manuscript no. processes-2709589 titled "Investigation on the impact of trailing edge stacking styles on hydraulic performance in the multistage submersible pump space diffuser"

These authors examined the effect of the wrap angles on the principle of corner separation vortex flow for the multistage submersible pump. Also they used a linear equation on the profile line of the diffuser trailing edge in order to optimize the wrap angle.

In my opinion, the abstract is a bit too extensive. Of course, this is interesting information, but only the most important information should be included in the abstract. So I suggest somehow condensing the fragment from lines 20 to 34 as much as possible.

Introduction

This important chapter is based on 22 sources. This is a sufficient number. MDPI Publishing House, as far as I know, prefers that references include abbreviated names of journals. So, for example, instead of "Journal of Fluids Engineering-transactions of The ASME" you should use J Fluids Eng J FLUID ENG-T ASME. Therefore, I rather suggest correcting all references. Generally, however, the sources used are rather young and come from good scientific journals. By the way, please correct some things in the introduction itself, such as "Goto A. [7-8]et al".

An additional and quite important comment regarding the introduction itself. It can be seen that the work done by these authors is rather large. However, the Review of the State of Knowledge does not convince the reader that the scientific achievements of these authors are important. I believe that despite having enough references, the introduction was not done very well. I suggest that these authors focus less on writing how many similar things have been done on this topic, but where is the gap that your research fills.

Methodology

This chapter needs to be improved drastically. Basically, I think that the solver in which the numerical mesh was made is the least important here. What is more important is the shape of the mesh itself, the mesh parameters for the boundary layer and the solver settings.

Comments on the Quality of English Language

I only suggest cosmetic language changes before resubmitting the revised paper.

Author Response

  1. In my opinion, the abstract is a bit too extensive. Of course, this is interesting information, but only the most important information should be included in the abstract. So I suggest somehow condensing the fragment from lines 20 to 34 as much as possible.

Response: Thanks for the reviewer’s valuable suggestions. The abstract has been condensed and expressed the important information. The modified section is shown in lines 23 to 27.

  1. This important chapter is based on 22 sources. This is a sufficient number. MDPI Publishing House, as far as I know, prefers that references include abbreviated names of journals. So, for example, instead of "Journal of Fluids Engineering-transactions of The ASME" you should use J Fluids Eng J FLUID ENG-T ASME. Therefore, I rather suggest correcting all references. Generally, however, the sources used are rather young and come from good scientific journals. By the way, please correct some things in the introduction itself, such as "Goto A. [7-8]et al".

An additional and quite important comment regarding the introduction itself. It can be seen that the work done by these authors is rather large. However, the Review of the State of Knowledge does not convince the reader that the scientific achievements of these authors are important. I believe that despite having enough references, the introduction was not done very well. I suggest that these authors focus less on writing how many similar things have been done on this topic, but where is the gap that your research fills.

Response: Thanks for the reviewer’s valuable advice. All references have been modified according to the MDPI Publishing House and corrected some errors. Some explanations have been added. The modified section is shown in lines 63-82 of the Introduction section.

  1. Methodology: This chapter needs to be improved drastically. Basically, I think that the solver in which the numerical mesh was made is the least important here. What is more important is the shape of the mesh itself, the mesh parameters for the boundary layer and the solver settings.

Response: Thanks for the reviewer’s valuable suggestions. The mesh of impeller and diffuser had been added in Figure 4.

Round 2

Reviewer 3 Report

Comments and Suggestions for Authors

I accept this paper as it is.